# Anatomical Development of the Cerebellothalamic Tract in Embryonic Mice

**DOI:** 10.3390/cells11233800

**Published:** 2022-11-27

**Authors:** Daniël B. Dumas, Simona V. Gornati, Youri Adolfs, Tomomi Shimogori, R. Jeroen Pasterkamp, Freek E. Hoebeek

**Affiliations:** 1Department of Neuroscience, Erasmus Medical Center, 3015 AA Rotterdam, The Netherlands; 2Department of Translational Neuroscience, MIND Facility, University Medical Center Utrecht Brain Center, University of Utrecht, 3584 CX Utrecht, The Netherlands; 3RIKEN Center for Brain Science, Laboratory for Molecular Mechanisms of Brain Development, Wako City 351-0198, Japan; 4Department for Developmental Origins of Disease, Wilhelmina Children’s Hospital and University Medical Center Utrecht Brain Center, 3584 EA Utrecht, The Netherlands

**Keywords:** cerebello-thalamic tract, embryonic development, cerebellum, thalamus, mouse

## Abstract

The main connection from cerebellum to cerebrum is formed by cerebellar nuclei axons that synapse in the thalamus. Apart from its role in coordinating sensorimotor integration in the adult brain, the cerebello-thalamic tract (CbT) has also been implicated in developmental disorders, such as autism spectrum disorders. Although the development of the cerebellum, thalamus and cerebral cortex have been studied, there is no detailed description of the ontogeny of the mammalian CbT. Here we investigated the development of the CbT at embryonic stages using transgenic *Ntsr1-Cre/Ai14* mice and in utero electroporation of wild type mice. Wide-field, confocal and 3D light-sheet microscopy of immunohistochemical stainings showed that CbT fibers arrive in the prethalamus between E14.5 and E15.5, but only invade the thalamus after E16.5. We quantified the spread of CbT fibers throughout the various thalamic nuclei and found that at E17.5 and E18.5 the ventrolateral, ventromedial and parafascicular nuclei, but also the mediodorsal and posterior complex, become increasingly innervated. Several CbT fiber varicosities express vesicular glutamate transporter type 2 at E18.5, indicating cerebello-thalamic synapses. Our results provide the first quantitative data on the developing murine CbT, which provides guidance for future investigations of the impact that cerebellum has on thalamo-cortical networks during development.

## 1. Introduction

Cerebello-cerebral connectivity is known to be involved in motor activity, but several non-motor functions are also controlled by long-range cerebellar and cerebral projection neurons [1,2,3,4,5]. The most direct route from the cerebellum to the cerebral cortex runs through the thalamic complex and it is this cerebello-thalamic tract (CbT) that has been implicated in a wide range of neurological conditions, such as rapid onset dystonia, epilepsy and autism spectrum disorder (ASD) [6,7,8,9]. Several of these pathologies have a developmental aspect, and thus put a focus on the ontogeny and maturation of the CbT. Imaging data from children with cerebellar lesions, which are at high-risk of developing ASD [10,11], revealed concomitant cerebral impairments that were suggested to be mediated by impairments of cerebello-thalamic connectivity. Despite recent technical advances that allow clinicians to investigate the developing cerebello-cerebral connectivity in very pre-term born children [12], it remains unknown from which age cerebellar axons reach the thalamic primordium and thus also from which age cerebellar abnormalities can influence thalamo-cortical development.

In animal models the development of cerebello-thalamic connectivity is equally understudied. Although it is well understood that cerebellar nuclei (CN) neurons are the sole source of the CbT, there are few studies available on the development of their axonal connections to the thalamic complex. Some reports on a developing CbT in the post-natal opossum [13], rat [14,15] and a single report in mouse embryo [16] qualitatively describe the timing of CN innervation of midbrain nuclei, such as the parabrachial and red nuclei, and the thalamic primordium, but no quantitative data on the ontogeny of cerebellar innervation of the thalamic complex are currently available. Given that early synaptic afferents have been shown to modulate thalamic activity patterns, gene expression profiles and the thalamo-cortical connectivity [17,18,19] is of importance to elucidate at what embryonic stage cerebellar axons start to innervate the developing thalamus. The cerebellum is especially important in this regard, given that in the adult rodent the CbT diverges to many first-order and higher-order nuclei, including ventrolateral (VL), ventromedial (VM), centrolateral (CL), posteriomedial (POm), parafascicular (Pf) and mediodorsal (MD) [20,21,22,23,24,25,26], which potentially have a critical period for growth and maturation of their afferents and efferents. For instance, the somatosensory ventrobasal nuclei receive brainstem afferents from E17.5 [27] and during the following three weeks thalamic axons initiate connections with various neuronal populations in the developing cerebral cortex [28], some of which are transient [29]. Disruptions in thalamic and cortical activity and connectivity result in abnormal efferent connections and functional aberrations (e.g., [30,31]). Thus, in the realm of thalamo-cortical connectivity developing activity-dependent and in a time-sensitive fashion, we set out to investigate the ontogeny of the cerebello-thalamic connectivity.

Here we investigated the embryonic development of the CbT using transgenic mice in which CbT fibers are tagged with red fluorescent protein (RFP). After confirming the presence of the CbT in wild type mice by in utero electroporation of CN neurons with a fluorescent tag, we used our transgenic mice to investigate the distribution of CN axons throughout the thalamic primordium between E15.5 and E18.5. Our results reveal that until E16.5 CN axons reside in the prethalamus and are found throughout several thalamic nuclei at E17.5. By E18.5 even the rostrally located VL nucleus is innervated by CN axons that can be labelled with a marker of active glutamatergic synapses.

## 2. Materials and Methods

All experiments were performed in accordance with the European Community’s Council Directive. Protocols were reviewed and approved by the Dutch national experimental animal committees (DEC) and every precaution was taken to minimize stress and the number of animals used in each series of experiments.

### 2.1. Mice

To visualize the fibers from the cerebellar nuclei (CN), mice carrying an *Ntsr1-Cre* allele [1] (Mutant Mouse Regional Resource Center; Stock *Tg(Ntsr1-cre)GN220Gsat/Mmucd*) were crossed with an *Ai14* reporter line [2] (Jackson laboratories strain 007908) to generate mice expressing RFP in Ntsr1^+^ cells. The lines were genotyped by PCR using dedicated primer sets (5′-GACGGCACGCCCCCCTTA-3′ for *Ntsr1* and 3′-AAGGGAGCTGCAGTGGAGTA-5′ and 5′-CCGAAAATCTGTGGGAAGTC-3′ for wildtype *Ai14* and 3′-CTGTTCCTGTACGGCATGG-5′ and 5′-GGCATTAAAGCAGCGTATCC-3′ for mutant *Ai14*). Only double heterozygous mutants were used for further studies. To investigate the prenatal development of the CbT fibers, we used *Ntsr1-Cre/Ai14* embryos aged from embryonic day (E) 14.5 to E18.5. The morning of the day of vaginal plug detection was counted as E0.5. Dams were deeply anesthetized with isoflurane before extracting the embryos. In total, we used four E14.5, two E15.5, five E16.5, seven E17.5 and six E18.5 embryos. For each age, the embryos were collected from at least two different mothers. There were no gross morphological abnormalities present in any of these embryos. We also used three adult mice (P48–75) for characterization of the *Ntsr1-Cre*-expressing CN neurons.

### 2.2. Tissue Preparation for Immunohistochemistry

Embryos were collected at E14.5, E15.5, E16.5, E17.5 and E18.5. Those of E14.5 and E15.5 were immediately immersion-fixed in 4% paraformaldehyde (PFA) in PBS; E16.5 and E17.5 were first decapitated before immersion fixation in 4% PFA; and E18.5 brains were immediately dissected in phosphate buffer saline (PBS) over ice before immersion fixation in 4% PFA. All embryo tissue was fixed for 36 h at 4 °C. After fixation, the embryos were cryoprotected in 20% sucrose for at least 3 days at 4 °C. After cryoprotection, the embryo tissue was embedded in 22% bovine serum albumin in 7% gelatin solution. The embedded brains were stored at −80 °C until sectioning. Sagittal and coronal sections (20 µm) were produced and glass-mounted on chrome alum-gelatin coated slides using a Microm HM560 cryostat (Walldorf, Germany) and then stored at −20 °C until processing for immunohistochemistry.

After anesthetizing the adult mice with 0.15 mL pentobarbital (intraperitoneal (i.p.) injection), they were perfused with 4% PFA. The brains of the adult mice were removed after perfusion and post-fixed on a shaker for 2 h in 4% PFA at room temperature. After perfusion, the dura mater was removed. Thereafter, the brains were embedded in 12% gelatin/10% sucrose. The gelatin blocks containing the brains were then incubated in 30% sucrose/0.1 M PB overnight at 4 °C. Thereafter, the brains were cut into 40 µm thick sections using a Leica SM 2000 R sliding microtome (Nussloch, Germany).

### 2.3. Immunohistochemistry

For 3,3-diaminobenzidine (DAB) staining, slides with embryonic sections were immersed in 0.3% H_2_O_2_ in methanol to block endogenous peroxidase activity. After three times of 10 min rinsing with PBS, the cell membranes were permeabilized by immersion in 0.3% Triton in PBS for 60 min. After another three times of 10 min rinsing with PBS, the slides were immersed in a 5% normal horse serum (NHS) in PBS blocking solution for 60 min. Afterwards, the slides were rinsed three times for 10 min with PBS and subsequently incubated in primary antibodies in 2% NHS/PBS overnight at 4 °C. After three times of 10 min rinsing with PBS, the slides were incubated in secondary antibodies in 2% NHS/PBS for 2 h at room temperature. After another three times of 10 min rinsing with PBS, the slides were incubated in avidin biotin complex (ABC) solution for 2 h. The ABC solution was prepared 40 min before incubation. The ABC solution consists of 0.7% avidin, 0.7% biotin and 0.5% Triton in PBS. After incubation in ABC solution, the slides were rinsed three times for 10 min with PBS and two times for 10 min with 0.05 M phosphate buffer (PB). After rinsing, slides were incubated for 15 min in DAB solution, which consisted of 0.665% DAB in 0.1 M PB. The DAB reaction was catalyzed by adding H_2_O_2_ to the solution (final concentration 0.01%) right before immersion. Afterwards, the slides were rinsed three times for 10 min in 0.05 M PB. After an additional short rinse in MilliQ, the slides were incubated in thionin for 5 min. Thereafter, the slides were incubated two times for 10 min in 96% ethanol, followed by three incubation steps of 2 min in 100% ethanol. Afterwards, the slides were incubated three times for 2 min in xylene and subsequently covered with Permount (SP15-500, Fisher Chemical, Waltham, MA, USA) and coverslipped.

To stain embryonic sections with immunofluorescence (see Table 1 and Table 2 for antibody information) the slides were first rinsed three times for 10 min with PBS and subsequently permeabilized by immersion in 0.3% Triton in PBS for 60 min. The slides were then incubated in 1% sodium dodecyl sulfate in PBS for 5 min to facilitate antigen retrieval. This was followed by three times 5 min rinsing with PBS. Thereafter, the slides were immersed in a 5% NHS/PBS blocking solution for 60 min. Afterwards, the slides were rinsed three times for 10 min with PBS and subsequently incubated in primary antibodies in 2% NHS/PBS overnight at 4 °C. After three times of 10 min rinsing with PBS, the slides were incubated in secondary antibodies in 2% NHS/PBS for 2 h at room temperature. After incubation, the slides were rinsed two times for 10 min with PBS and 10 min with 0.05 M PB. The slides were then incubated in 1:10,000 4’,6-diamidino-2-fenylindool (DAPI) solution for the visualization of cell nuclei. Afterwards, the slides were rinsed two times for 10 min with 0.05 M PB and subsequently covered with Mowiol (4-88; Sigma Aldrich, Zwijndrecht, The Netherlands) and coverslipped.

For immunofluorescence on adult tissue, the adult brain sections were first rinsed four times for 10 min with PBS and subsequently preincubated in 10% NHS/0.5% Triton/PBS for 60 min. This was followed by incubation in primary antibodies in 2% NHS/0.4% Triton/PBS for 48 h at 4 °C. Afterwards, the sections were rinsed four times for 10 min with PBS and subsequently incubated in secondary antibodies in 2% NHS/0.4% Triton/PBS for 90 min at room temperature. The sections were then rinsed two times for 10 min with PBS and 5 min with 0.1 M PB. After rinsing, the sections were incubated in 1:10,000 DAPI solution for 10 min. The sections were then rinsed two times for 5 min with 0.1 M PB, after which they were immersed in 5% gelatin/1% chrome alum/MilliQ. Afterwards, the sections were glass-mounted and coverslipped using Mowiol. Except for the incubation in primary antibody solution, all the steps were performed at room temperature.

### 2.4. DISCO

For this procedure, embryos were collected at E15.5, E16.5, E17.5 and E18.5. The brains of E16.5–18.5 embryos were immediately dissected in phosphate buffer saline (PBS) over ice before immersion fixation in 4% PFA. Younger animals were decapitated and their heads were immediately immersion-fixated in 4% PFA. Brains were fixed for 24 h at 4 °C and then incubated in 0.2% gelatin/0.5% Triton/PBS (PBSGT) for 24 h on a shaker (~70 rounds per min (rpm)) at room temperature. The PBSGT was filtered with a 0.2 µm filter before use. After incubation in PBSGT, brains were incubated in a 0.2 µm filtered primary antibody solution in 0.1% saponin (Sigma Aldrich S-7900)/PBSGT at 37 °C on a shaker (~100 rpm) for 1 week. Afterwards, brains were rinsed six times for 60 min with PBSGT and subsequently incubated overnight in a 0.2 µm filtered secondary antibody solution in 0.1% saponin/PBSGT at 37 °C on a shaker (~100 rpm). Afterwards, brains were rinsed six times for 60 min with PBSGT. Then brains were incubated in 50% tetrahydrofuran (THF; including 250 ppm butylated hydroxytoluene as a stabilizer) (Sigma Aldrich 186562-1L) in H_2_O overnight to start dehydration. Thereafter, brains were incubated for 60 min in 80% THF/H_2_O, then two times for 60 min in 100% THF. Brains were then incubated in dichloromethane (Sigma Aldrich 270997-1L) for 20 min for clearing the brains, after which they were incubated and stored in dibenzylether (Sigma Aldrich 108014-1KG) at room temperature. From the first THF incubation step onwards, care was taken to have the least amount of air possible in the vials containing the brains (see also [3]).

### 2.5. In Utero Electroporation

In utero electroporation (IUE) was performed as described [4]. Briefly, timed-pregnant ICR mice with E10.5 embryos were anesthetized with pharmaceutical grade sodium pentobarbital (50 μg per gram body weight). After incision at the abdominal midline, the uterine horns were carefully placed onto a 37 °C prewarmed PBS-moistened cotton gauze. A flexible fiber optic cable was placed under the uterine horn for visualization of the embryo. After positioning the embryo, a glass capillary was carefully inserted into the fourth ventricle. Embryos were injected with 1 μL of pCAG-EYFP plasmid DNA solution (prepared in TE; 10 mM Tris base, 1 mM ethylenediaminetetraacetic acid solution, pH 8.0), 0.1% Fast Green. Embryos were electroporated by inserting a custom-made fine tungsten negative electrode into the fourth ventricle and a custom-made platinum positive electrode into the uterus, placing one hemisphere of the cerebellum between the two electrodes (see [4] for preparation of electrodes). Three square-wave pulses (7V, 100 ms duration) were then delivered with a pulse generator (A-M Systems Model 2100, Sequim, WA, USA). The uterine horns were returned into the abdominal cavity, the wall and skin were sutured, and the embryos were allowed to continue their normal development until they were sacrificed at E18.5.

### 2.6. Antibodies

An overview of the antibodies used is presented in Table 1 and Table 2. For DAB staining, we used a primary rabbit anti-RFP antibody (1:1000, Rockland Immunochemicals, Limerick, PA, USA) and a secondary donkey anti-rabbit antibody (1:200, Jackson, Cambridge, UK). For immunofluorescence, we used chicken anti-calbindin (1:500, Synaptic Systems, Goettingen, Germany), mouse anti-calbindinD-28 (1:500, Sigma-Aldrich), chick anti-GFP (1:200, Abcam, Cambridge, UK), goat anti-FoxP2 (1:500, Santa Cruz, Santa Cruz, CA, USA), guinea pig anti-vGluT2 (1:500, MilliPore, Amsterdam, The Netherlands), rabbit anti-RFP (1:1000, Rockland) and mouse anti-NeuN (1:1000, MilliPore) as primary antibodies. Cy5 anti-chicken (1:200, Jackson), Cy3 anti-mouse, FITC anti-chick, (1:200, MilliPore), Alexa488 anti-goat (1:200, Jackson), Cy5 anti-guinea pig (1:200, Jackson), Cy3 anti-rabbit (1:400, Jackson) and Alexa488 anti-Mouse (1:200, Jackson) antibodies were used as secondary antibodies.

For 3DISCO, we used chicken anti-calbindin (1:500, Synaptic Systems), goat anti-FoxP2 (1:500, Santa Cruz) and rabbit anti-RFP (1:000, Rockland) as primary antibodies. Cy5 anti-chicken (1:200, Jackson), Alexa488 anti-goat (1:200, Jackson) and Cy3 anti-rabbit (1:400, Jackson) antibodies were used as secondary antibodies.

### 2.7. Imaging

An overview of the microscopes used is presented in Table 3. Light microscopy pictures of DAB-stained slides were made using a Nanozoomer 2.0-HT (Hamamatsu, Japan) at 40× magnification captured using a 20× objective with an NA of 0.75. The pixel size was 230 nm × 460 nm. Confocal microscopy pictures were taken with a Zeiss LSM700 Meta (Carl Zeiss Microscopy, Breda, Netherlands), an Olympus FV3000 (Olympus, Tokyo, Japan) and an Opera Phenix HCS system (Perkin Elmer, Hamburg, Germany). Confocal pictures on the Zeiss LSM700 Meta were taken with a 63× oil Plan-Apochromat lens with an NA of 1.4. Z-stacks were taken with a voxel size of 50 × 50 × 150 nm, a pinhole of 1 Airy unit and bit depth of 8-bits. Signal-to-noise ratio was improved by 4× line averaging. For the different fluorophores, the following lasers were used: 405 nm for DAPI, 488 nm for Alexa488, 543 nm for Cy3 and 633 nm for Cy5. The confocal pictures IUE tissue were acquired on a Keyence (BZ-X800, Itasca, IL, USA) and taken with a 10× lens with an NA of 0.45. The confocal pictures on the Opera Phenix HCS system were taken with a 20× lens with an NA of 0.4. The pixel size was 598 nm × 598 nm. The bit depth was 16-bits. For the different fluorophores, the following lasers were used: 488 nm for Alexa488, 561 nm for Cy3 and 640 nm for Cy5. Z-stacks consisting of two 10 µm spaced slices were taken to correct for shifts in the z-axis. Before further analysis, a maximum intensity projection of the images was produced, which was subsequently converted to 8-bits.

The 3DISCO cleared brains were imaged using LaVision biotec light sheet microscope Ultramicroscope II (LaVision biotec, Bielefeld, Germany). The microscope consists of an Olympus MVX-10 Zoom Body (0.63–6.3×) equipped with an Olympus MVPLAPO 2× objective lens, which includes dipping cap with 6 mm working distance. Images were taken with a Neo sCMOS camera (Andor) (2560 × 2160 pixels; pixel size: 6.5 × 6.5 µm). Samples were scanned with two-sided illumination, a sheet NA of 0.148348, which results in a 5-µm thick sheet. We applied a step-size of 2.5 µm using the horizontal focusing light-sheet scanning method and contrast blending algorithm. The effective magnification for overview pictures was 3.2× and for detailed images 12.6×. The lens had an NA of 0.5 and bit depth was 16-bits. Overview pictures had a voxel size of 2030 nm × 2030 nm × 2500 nm and detailed pictures had a voxel size of 515.9 nm × 515.9 nm × 2500 nm. The following laser filter combination was used to image Cy3: Coherent OBIS 561-100 LS Laser with 615/40 filter.

### 2.8. Delineation and Intensity Measurements

Any images containing structural artefacts (e.g., freezing artifacts and cutting artifacts) were discarded. The delineation of anatomical regions and measurements were performed with FIJI software (version 1.51). To delineate the thalamic nuclei across the different ages studied, we used the chemoarchitectonic atlas of the developing mouse brain [5], the atlas of the prenatal mouse brain [6], the atlas of the developing mouse brain [7], the Allen brain atlas [8] and descriptive studies of FoxP2 expression in developing mice [8,9,10,11]. With FoxP2, we could delineate Pf, MD, VB, POm, VM and midline (ML) nuclei, the latter of which we defined as centromedial, paraventricular, intermediodorsal, reunions, retroreuniens, intermediodorsal, rhomboid, xyphoid, retroxyphoid, interanteromedial, anteromedial and posteromedian nuclei of the thalamus. When combining FoxP2 with calbindin staining, we could delineate the VM and the ML more specifically than with FoxP2 alone [12]. The border between VL and the intralaminar nuclei could not consistently be accurately delineated in all slices. Therefore, the size of the nucleus might be underestimated in some instances. Moreover, at E16.5, the border between the VL and the LP could not be delineated. We therefore did not quantify the VL at this age. The delineation of thalamic nuclei was performed in the fluorescent channels representing the FoxP2 and calbindin expression, in order that the researcher was blinded for the location of RFP signal, which represents the location of *Ntsr-Cre/Ai14*-positive axons. For further measurements, nuclei were delineated with the ROI manager in FIJI. If a nucleus could not be delineated in two or more sequential sections, the data for this particular nucleus of that mouse were excluded.

To measure the amount of Cy3 signal in each nucleus, the mean plus two times the standard deviation of the histogram was used as threshold to binarize the image into background and foreground. To measure the area of detected objects within each nucleus (A_detected_) we used the ‘analyze particle’ function.

### 2.9. Colocalization and Volume Measurement

Colocalization of vGluT2 and RFP signals was determined using FIJI in the 63× z-stacks of the different thalamic nuclei. After applying user defined thresholds in the separate channels of the raw z-stacks, sites of putative localization were subjectively identified by the appearance of structures showing the presence of both colors and then deconvolved using Huygens (Scientific Volume Imaging). After deconvolution, a user defined threshold was applied to confirm the colocalization, which was defined as a complete overlap of vGluT2 and RFP structures (see also [13]. In short, vGluT2 positive terminals’ volume in VL and VM was measured using a custom-written FIJI macro. This macro used manual thresholding and particle analysis to acquire masks of the vGluT2 and RFP channels on which the Image Calculator was used to acquire an image showing only the overlap between vGluT2 and RFP. The volume thereof was then determined with FIJI’s 3D object counter.

### 2.10. Statistical Analysis

The fluorescence of a whole thalamic nucleus was calculated for each mouse by summing the A_detected_ and the A_delineated_ of all the sections containing the nucleus in question, yielding the ∑A_detected_ and ∑A_delineated_, respectively, per nucleus per hemisphere. Dividing ∑A_detected_ by ∑A_delineated_ results in the sum of the relative area occupied by RFP^+^ fibers, expressed by the percentage of summed area occupied (pSAO). For each mouse, data from both hemispheres were averaged. Thereafter, the average pSAO per nucleus was calculated for E16.5, E17.5 and E18.5. For each nucleus, significant differences between the ages were first tested with a Kruskal–Wallis (K-W) test (degrees of freedom = 2). When a significant difference was found, Dunn’s post hoc test was used to compare pairwise between the age groups. For the latter test, a Šidák corrected *p*-value of 0.0127 was used as threshold for significance. To compare the data of nuclei gathered from only two ages we used a Mann–Whitney U test with a Šidák corrected *p*-value of 0.017 as threshold for significance. To compare the data of MD, POm and VB combined with VM, VL and Pf separately, we used a Mann–Whitney U test with a Šidák corrected *p*-value of 0.0127. Šidák corrected *p*-values are calculated using $\alpha_{per\;comparison}=1-{\left(1-\bar{\alpha }\right)}^{1/k}$, where $\bar{\alpha }$ is the overall significance level, which was chosen to be 0.05, and k was chosen as the amount of times the same dataset was analyzed, or the amount of comparisons in the case of a Dunn’s post hoc test.

The relative amount of RFP^+^ fibers per section was calculated by dividing the A_detected_ by the A_delineated_. These values were then plotted against the relative caudal-to-rostral distance. This distance was calculated as a linear scale from 0 to 100, with 0 indicating the most caudal section in which a particular nucleus was delineated and 100 indicating the most rostral section in which that nucleus was delineated. To determine whether there was caudal-to-rostral gradient of the relative amount of RFP^+^ fibers, a Spearman’s rank correlation coefficient ‘rho’ was calculated per nucleus per age. To measure differences in correlation between groups, the Fisher’s Z-transformation was used, after which a Z-test was conducted for pairwise comparison. Since the same dataset was used for the former analysis, a Šidák corrected *p*-value of 0.017 was used as threshold for significance.

## 3. Results

### 3.1. Proportion of CN Neurons Labelled RFP^+^ in Ntsr1-Cre/AI14 Mice

To study the embryonic development of the CbT, we crossed the *Ntsr1-Cre* and *Ai14* mouse lines. Offspring heterozygous for both constructs is characterized by RFP^+^ expression in previously characterized large diameter CN neurons (Figure 1A–D) (see also [14]). To assess the distribution of Cre expression in the cerebellum we quantified the proportion of RFP^+^ CN cells and the number of NeuN^+^ cells in the CN. Since there are non-CN neurons migrating through the subcortical part of the cerebellum at embryonic stages [15,16], we quantified the RFP^+^ CN population in adult mice (Figure 1, Table 4) under the assumption that the ratio of RFP^+^ neurons in the lateral cerebellar nuclei, interposed nucleus and medial cerebellar nucleus does not change between late embryonic and adult stages. We based this assumption on data showing that the proliferation of excitatory CN neurons is completed at E12.5 [32], i.e., well before our in-depth analysis of CbT. Our analyses of adult stages show that most of these RFP^+^ neurons reside in the interposed nucleus and appeared larger than RFP^-^ somata. In contrast to the CN, no RFP^+^-neurons were found in the nearby vestibular nuclei or in the cerebellar cortex (data not shown).

### 3.2. Ontogeny of Cerebello-Thalamic Connection

To determine at which embryonic age RFP^+^ CN axons arrive at the thalamic primordium, we used a combination of light-sheet imaging of 3DISCO-treated brains and confocal and light-microscopy of histological sections. At E14.5 the CbT did not yet reach diencephalic structures (data not shown; see also [17]). At E15.5 and E16.5 the CbT is positioned dorsally in the mesencephalic curvature and has progressed beyond the red nucleus to reach the prethalamus (Figure 2). Yet, in E16.5 brains we found no evidence for CbT fibers invading the thalamic complex.

### 3.3. Invasion of Thalamic Anlage by Putative CbT Axons from E17.5

From E17.5 the CbT commences to innervate the thalamic nuclei (Figure 3). Using light-sheet microscopy of fluorescently labelled neuronal structures (Figure 3A) and conventional light microscopy of DAB-stained RFP^+^ axons (Figure 3B–E) we found that at this stage putative CbT axons are invading the thalamic anlage. To establish which thalamic nuclei are invaded by RFP^+^ axons we combined immunofluorescent staining for FoxP2 and calbindin-d28K on alternating slices, combined with RFP-staining to identify the neuronal populations that form the separate nuclei (Figure 4A for E18.5). We found that the bundle of RFP^+^ axons extended from the prethalamus into the nearby VM nucleus and diverged towards other nuclei (Figure 4B,C).

### 3.4. Extracerebellar RFP^+^ Neurons and Axons

We noted that during embryonic development sparsely distributed RFP^+^ neurons are also found outside of the CN (Figure 5). We found RFP^+^ neurons throughout various regions: the posterior and lateral hypothalamus, hippocampus, lateral geniculate nucleus, medial reticular formation, superior colliculus and the Rim, i.e., nuclei that primarily project to regions other than the cerebellar-receiving thalamic nuclei. The only other region that showed pronounced RFP^+^ expression at E18.5 and is known to project to cerebellar-receiving thalamic nuclei was layer VI of the cerebral cortex [1,18]. Since cortical layer V and VI neurons innervate the thalamic complex from E17.5 onwards [19], we investigated the relative location of their axons compared to RFP^+^ axons from CN neurons. We found that at E18.5 RFP^+^ axons from cortical layer VI via the internal capsule reached the VB nucleus, but remained lateral of VM (Figure 6). The RFP^+^ CN axons reside more medially in a bundle that enters the thalamic complex from the mesencephalic and subthalamic regions.

### 3.5. Cerebellar In Utero Electroporation Confirms CbT Position at E18.5

To confirm the presence of cerebellar axons in the thalamic primordium at E18.5, we labelled CbT fibers using IUE transfection with an EYFP-construct at E10.5. Apart from a dense population of transfected cerebellar neurons and sparse labelling in the brainstem (Figure 7A,B), the rest of these E18.5 brains solely contained labelled axons with dense diencephalic projections. Qualitatively, the CbT labelled by IUE transfection is similar to the presumptive CbT labelled in our *Ntsr1-Cre/Ai14* embryos at E18.5 (Figure 7C), in that ventrobasal thalamic nuclei are largely avoided by CN axons in contrast to, for instance, VM. These results from IUE transfection of wildtype hindbrain neurons indicate that at late embryonic stages RFP^+^ axons in *Ntsr1-Cre/Ai14* thalami predominantly represent CbT fibers, which together support the notion that CbT fibers start innervating thalamic nuclei around the same time as corticothalamic layer VI.

### 3.6. Quantification of RFP^+^ Axons in Thalamic Nuclei Indicates Increasing CbT Density

To quantify the innervation of the individual nuclei by RFP^+^ axons, we selected all sections available from *Ntsr1-Cre/Ai14* for the identified nuclei and summed the percentage of the area that was RFP^+^ (see also Table 5). At E17.5 the surface covered by VM, VL and Pf, i.e., nuclei which receive CN innervation in the adult mouse brain [13,20], was ≥1% RFP^+^ (VM: 4.85 ± 4.13%; VL: 2.20 ± 1.41%; Pf: 1.06 ± 0.41%) (Figure 8). In E18.5 tissue the surface of RFP-signal further increased: in VM we found 14.7 ± 2.1% of the section’s surface to be RFP^+^, in VL 9.56 ± 2.95% and in Pf 3.30 ± 1.55% (Figure 9A–C). We found that during the final days of embryonic development the relative surface of RFP^+^ signal within MD and POm increased (Figure 9D–G). At E18.5 the MD, POm and VB did not differ significantly from one another, but compared to these nuclei the surface was found to be significantly less in ML. In addition, the relative surface of RFP-signal within MD, POm and VB at E18.5 combined (1.08 ± 0.54%) remained below the RFP fluorescence levels found in VM (*p* = 0.0036) and VL (*p* = 0.0036) but not Pf (*p* = 0.015).

To describe the development of the CbT in more detail, we focused on the VM, VL and Pf nuclei and analyzed the caudal-to-rostral gradient of the relative amount of RFP^+^ fibers (Figure 10). We calculated the Spearman’s correlation value rho (r) for E17.5 and E18.5 tissue pooled from various embryos (see methods) and found that in all these nuclei the fluorescence was relatively higher in most caudal sections. In VM at E17.5 we found significant negative correlation between the rostro-caudal level and the relative amount of RFP^+^ fibers (r = −0.550, df = 48, *p* = 3.57 × 10^−5^) (Figure 10A). Furthermore, in E18.5 embryos we found this negative correlation value for VM (pooled r = −0.790, df = 57, *p* = 1.06 × 10^−13^) (Figure 10B). When comparing the Fisher’s Z-transform of the correlation values for VM, we found that the correlation was significantly stronger at E18.5 than at E17.5 (−0.790 vs. −0.550, Z = −2.29, *p* = 0.0226; for E18.5 vs. E17.5, respectively). In VL at E17.5 there was a negative correlation between the rostro-caudal level and the relative amount of RFP^+^ fibers (r = −0.711, df = 44, *p* = 3.11 × 10^−8^) (Figure 10C). At E18.5 this correlation seemed weaker (r = −0.591, df = 50, *p* = 4.06^−6^) (Figure 10D); however, when comparing the Fisher’s Z-transform of the correlation values this correlation did not differ significantly from that of the E17.5 animals (Z = 1.01, *p* = 0.844). In the Pf we found the same correlation at E17.5, (r = −0.658, df = 39, *p* = 2.99 × 10^−6^) (Figure 10E), but not at E18.5 (pooled r = −0.265, df = 39, *p* = 0.0946) (Figure 10F), resulting in a significantly stronger correlation at E17.5 than at E18.5 (Z = −2.26, *p* = 0.0244).

Thus far, we identified the location of RFP^+^ axons and the growth of the CbT in the dorsal thalamic complex. We next evaluated whether RFP^+^ axons formed synaptic contacts using confocal microscopy and FoxP2-stained tissue (Figure 6 and Figure 11A). Whereas our DAB and immunofluorescent staining of E17.5 tissue did not provide any hint for RFP^+^ bouton-like varicosities in thalamus (data not shown), at E18.5 RFP^+^ axons show morphological characteristics of presynaptic terminal formation throughout the thalamic complex (Figure 11B–I). To confirm the subcortical origin of these terminals we co-labelled sections with vGluT2, i.e., a synaptic marker that separates subcortical axons from vGluT1-positive cortical layer VI axons [21,22,23]. Stacks of deconvolved, high-magnification confocal images of VL (Figure 11B–E) and VM (Figure 11F–I) tissue confirmed the colocalization of vGluT2^+^ and RFP^+^ putative axon terminals. We found that the vGluT2-labelled RFP^+^ boutons in VL appear to have a larger volume (VL: 1.48 ± 0.68 µm^3^, n = 7 terminals; VM: 0.58 ± 0.57 µm^3^, n = 7 terminals; *p* < 0.05, Mann–Whitney test) (Figure 11J).

## 4. Discussion

We describe the outgrowth of CN axons into the thalamic complex. Our data reveal that at E15.5 and E16.5 the CbT reached the prethalamus and from E17.5 onwards is detected throughout the developing thalamus. We found that for most of the investigated thalamic nuclei the RFP^+^ innervation increased significantly between E17.5 and E18.5. Moreover, for VL and VM the innervation also commenced rostrally. We also found that at E18.5 RFP^+^ axon varicosities colocalized with the presynaptic marker vGluT2 and that the average volume in VL was larger than in VM.

### 4.1. Technical Considerations

Using *Ntsr1-Cre/Ai14* brains for our current study allowed us to investigate the development of CbT axons, but also resulted in RFP^+^ expression in other cell populations, some of which are known to innervate the thalamic complex. Apart from the CN labelling (see also [14]), RFP^+^ neurons and axons were also readily identified in the deeper layers of the cerebral cortex, where *Ntsr1* is known to be expressed by a subpopulation of L6 pyramidal cells [1,18]. As we have shown in our analysis, the corticothalamic (CT) tract, which contains the L6 fibers, and the CbT are positioned differently in the embryonic mouse brain (Figure 3). Furthermore, the time of thalamic invasion by the CT and CbT fibers is different: our data reveal that from E17.5 RFP^+^ fibers already start invading the thalamic complex, which precedes the innervation of sensory and motor thalamic nuclei by CT fibers, which has been shown to occur earliest from E18.5 (as reviewed by [24]). Finally, the nuclei that are innervated by CT and cerebello-thalamic tracts differ at E18.5; i.e., CT fibers initially innervate the ventrobasal nuclei and appear to diverge from there onwards [19], whereas the CbT initially innervates the ventromedial nucleus.

We also found sparse labelling of somata in several other brain regions, which has implications for the interpretability of our data on thalamic innervation in the embryonic stages. Of these regions, we found that mRF cells that project to the thalamus are located more dorsally than where we observed Ntsr1^+^ cells [25]. We also located Ntsr1^+^ cells in the lateral and posterior hypothalamus, which in principle could also be a source of RFP^+^ axons in the studied thalamic nuclei. However, the primary projection targets of lateral and posterior hypothalamus projection neurons are located in the thalamic midline [26,27], which in our mouse model has the lowest RFP-signal. Since the projection patterns of electroporated GFP^+^ CN neurons appeared similar to the RFP^+^ projection patterns, we conclude that the proportion of RFP^+^ fibers of non-cerebellar origin in the VL and VM nuclei is limited.

### 4.2. Development of the CbT in Mouse Embryos

In the present study, we investigated the anatomical development of the CbT in mouse embryos aged E15.5–E18.5 (Figure 12) and found that prior to the entry of the thalamus, the CbT appears to stall its growth and reside in the prethalamus at least until E16.5. Several other thalamic afferents reveal such a ‘pause’ or waiting period, such as the trigeminal projection to the sensory VB nucleus [28] and CT pathways [19]. For the CbT this waiting period could come about due to several reasons: (i) to gain axonal energy supplies to allow axonal branching which could demand increased mitochondria production [29]; (ii) although most of the synaptic targets of CN axons are already born, CbT fiber invasion of the thalamus could be halted until the neuronal migration in the cerebellum nears completion [30] and (iii) CbT fibers, which originate from CN neurons born sequentially [15], might regroup in the prethalamus before entering the thalamic complex—a phenomenon described for the CT tract [24]. Furthermore, the mechanisms that facilitate the CbT waiting period remains to be elucidated. We hypothesize that the CbT is stalled temporarily by chemical signaling. Possible candidate signaling cascades are semaphorin-based, [31,33], albeit that numerous other molecules and receptors can potentially also influence the growth of CbT axons [34].

Upon entry into the thalamus, CbT fibers appeared to swiftly locate the VM and VL nuclei, which at adult stages are also their prime target nuclei [13,35,36]. We found that the most caudal portions of VM and VL nuclei showed more RFP^+^ labeling than the rostral portions, indicating that the afferents arise from caudal, which matches with the position of the CbT. The fact that the rostro-caudal gradient disappeared in Pf could suggest that CbT innervations to Pf have already stabilized at E18.5. Further investigations of the various thalamic afferents shall reveal more insights into how their growth is organized.

### 4.3. Timing of Invasion in the Thalamus

The thalamic invasion by CbT fibers starts from approximately E17.5 and most likely continues well into the postnatal period. This is relatively late when compared to other subcortical thalamic afferents. Both primary sensory afferents and serotonergic afferents start invading the thalamus prior to CbT fibers. The retinogeniculate pathway is present in the LG as early as E15.5 in mice [37] and the trigeminothalamic pathway invades the thalamus between E14 and E17 in mice [28,38]. Serotonergic afferents from the brainstem enter the thalamus at approximately E16 in rats [39]. As has been proposed before based on studies of the opossum, these ascending brainstem fibers may provide a passage for the CbT fibers through the mesencephalon [40]. Further research into the developing murine brain hall reveal the potential importance of the growth of ascending brainstem axons and CbT fibers.

It remains speculative what role the formation of cerebello-thalamic connectivity during late embryonic stages has, not in the least because the cerebellar cortex develops relatively late and its impact on the CN neurons is rudimentary in the antenatal period [41]. One option is that CbT fibers are present in the thalamic complex to interact with thalamic neurons and guide their axonal growth. The invasion of thalamo-cortical (TC) fibers into the cortical plate starts from E18.5–P0.5, but already before that the activity levels of thalamic neurons are thought to direct the axonal growth speed [37,42,43]. Thus, since the CbT seems to have established connections with the thalamus starting at E18.5, there is ample time for the cerebellar output to affect TC development. In addition, our data reveal a difference in vGluT2^+^ RFP^+^ CN axonal bouton size between VL and VM at E18.5. Whether a difference in axonal bouton volume reflects a difference in synaptic strength at E18.5 is unknown. In the adult mouse the CN axonal bouton volume has been reported to not be different between the VL and VM nuclei [13] and thus it remains to be investigated from which developmental stage this bouton volume in VL and VM equalizes.

Both the TC system and the cerebellum have been implicated to undergo critical periods [28,44,45]. Disruptions of the cerebellar development are suggested to cause disrupted behavioral traits in the motor and non-motor domain, linked to various neurodevelopmental disorders and psychiatric diseases, such as autism spectrum disorder and schizophrenia [36,46]. These neurological conditions are thought to be related to malformations in the neuronal wiring caused by adverse early life events [47,48] and our current study suggests that in the murine brain cerebellar aberrations could possibly derail thalamocortical wiring from E17.5 onwards, which translates to early fetal stages in human development. Future experiments can investigate the impact of cerebellar abnormalities on TC development and cortical maturation.

For a list of the abbreviations used in the main text, please refer to Appendix A.

## Figures and Tables

**Figure 1 cells-11-03800-f001:**
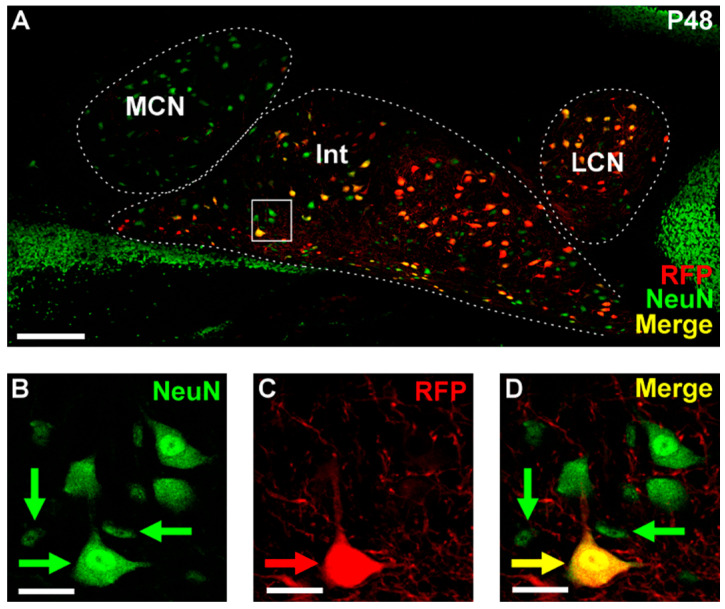
Example of NeuN and RFP staining in adult CN. (**A**) 20× tilescan of a coronal P48 cerebellar slice, zoomed in on the left CN. Stainings: red fluorescent protein (RFP) in red, NeuN in green, yellow indicating colocalization of these two stainings. (**B**–**D**) Zoom in of boxed region in (**A**). (**B**) NeuN stained cells. (**C**) RFP-stained cells and fibers. (**D**) Merge of B and C. Note that the RFP^+^ neurons are relatively big and all small neurons are RFP^−^. A bar plot representation of the quantification of RFP^+^ cells as a proportion of NeuN^+^ cells (N = 3 mice). Scale bars: (**A**) = 150 µm, (**B**–**D**) = 25 µm. CN = cerebellar nuclei, MCN = medial cerebellar nucleus, Int = interposed nucleus, LCN = lateral cerebellar nucleus.

**Figure 2 cells-11-03800-f002:**
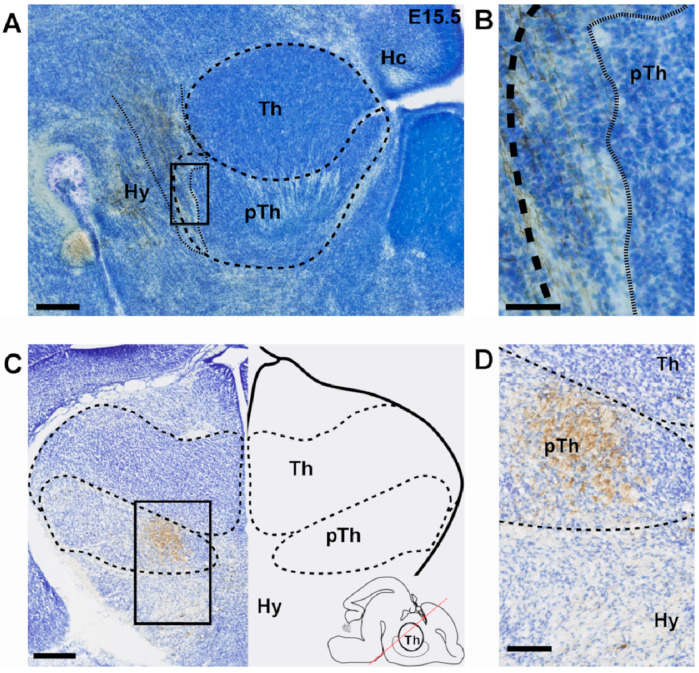
RFP^+^ fibers in the prethalamus at E15.5 and E16.5. (**A**,**B**) A sagittal section at E15.5 showing the presence of DAB-stained RFP^+^ fibers in the pTh and absence of RFP^+^ fibers in the thalamus at this age. Counterstained with thionin. Finer dotted line outlines RFP^+^ fibers. (**C**,**D**) Coronal section of an E16.5 mouse brain with inset, respectively, showing presence of DAB-stained RFP^+^ fibers in the pTh and absence of RFP^+^ fibers in the thalamus at this age. Counterstained with thionin. Inset in (**C**) indicating the rostro-caudal level and angle of the coronal slice with the red line. Scale bars: (**A**) = 100 µm, (**B**) = 50 µm, (**C**) = 250 µm, (**D**) = 100 µm. Hy = hypothalamus, Th = thalamus, pTh = prethalamus.

**Figure 3 cells-11-03800-f003:**
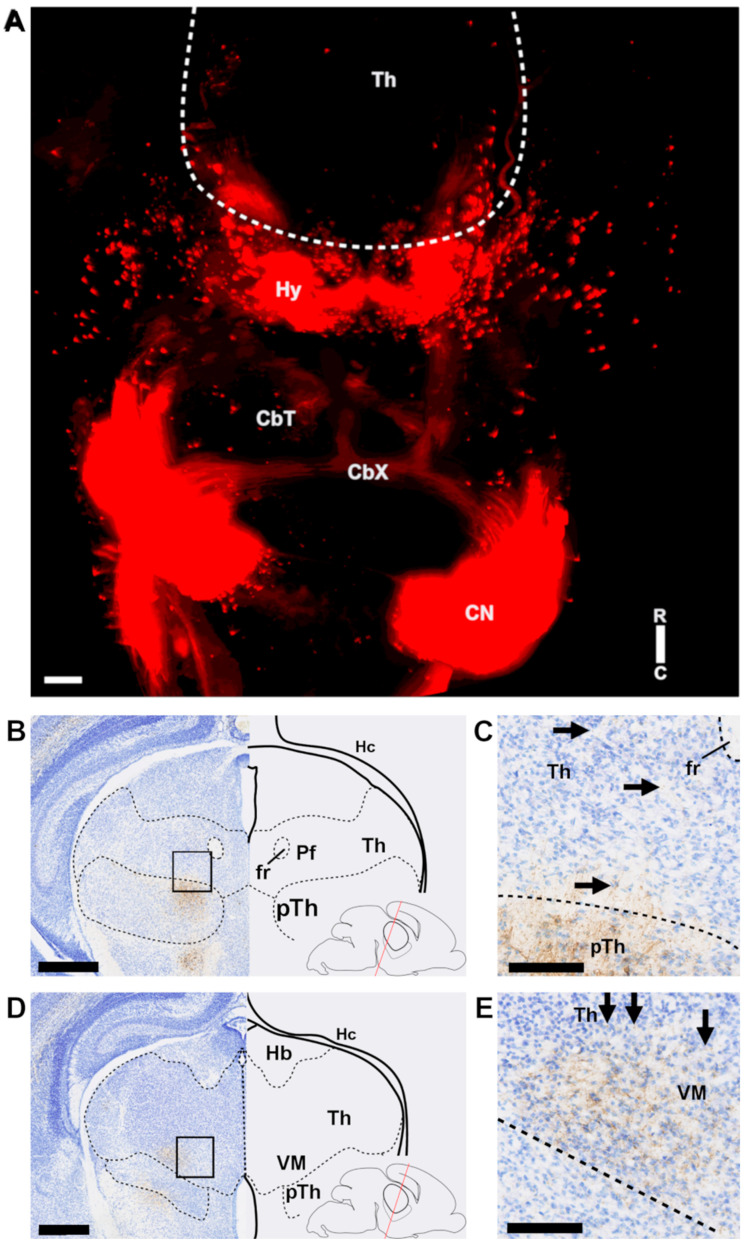
RFP^+^ fibers in the diencephalon at E17.5. (**A**) Horizontal view of a maximum intensity projection of a 3DISCO cleared E17.5 mouse brain. Compass in bottom right indicating directions of rostral (R) and caudal (C) sides. (**B**–**E**) Coronal section of an E17.5 mouse brain, where (**C**,**E**) are zoomed in on the boxes in (**B**) and (**D**), respectively. In the bottom right of (**B**,**D**), schematic showing the angle and rostro-caudal level of the coronal slices. Fibers can be seen in the Th, though they are much less bundled than in the medial pTh. Scale bars: (**A**) = 100 µm; (**B**,**D**) = 250 µm; (**C**,**E**) = 100 µm. CbT = cerebellothalamic tract, CbX = decussation of the cerebellar tract, CN = cerebellar nuclei, fr = fasciculus retroflexus, Th = thalamus, pTh = prethalamus, Hc = hippocampus, Hb = habenula, Pf = parafascicular nucleus, VM = ventromedial nucleus.

**Figure 4 cells-11-03800-f004:**
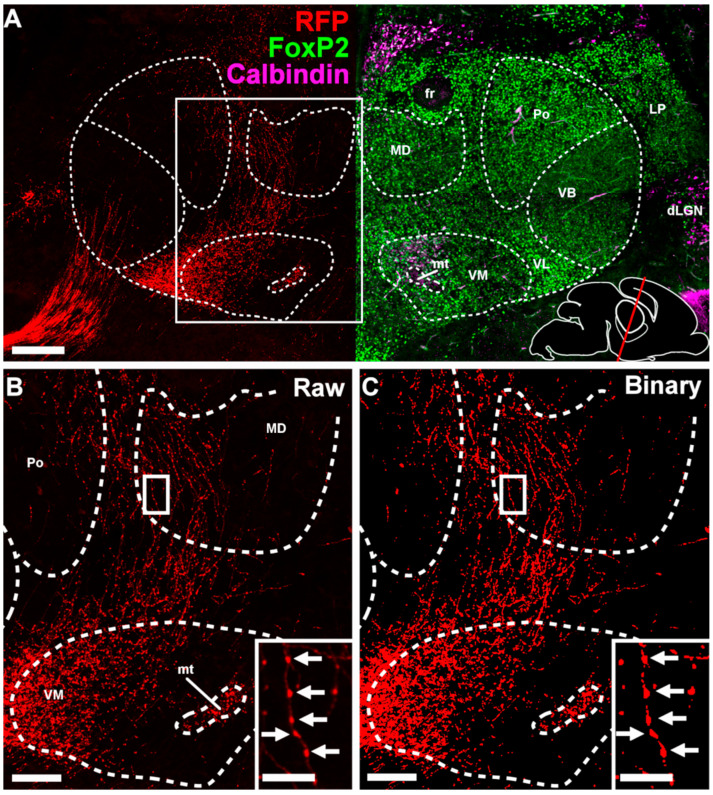
RFP^+^ fibers in the diencephalon at E18.5. (**A**) Coronal section of an E18.5 mouse brain stained for FoxP2 (in green), calbindin (magenta) and Red Fluorescent Protein (in red) showing an example of delineation of the ventromedial (VM), posterior (POm), mediodorsal (MD), ventrobasal (VB), ventrolateral (VL), lateral posterior (LP) and dorsal lateral geniculate (dLGN) nuclei, the fasciculus retroflexus (fr) and the mamillothalamic tract (mt). Bottom right shows a schematic showing the angle and rostro-caudal level of the coronal section. (**B**) Zoom in of boxed region in (**A**). In the inset, a single axon with multiple putative boutons is visible. Arrows indicate the putative boutons. (**C**) Binarized version of B after applying the threshold. Inset is a binarized version of the inset in C. Scale bars: (**A**) = 200 µm; (**B**,**C**) = 100 µm, insets = 25 µm.

**Figure 5 cells-11-03800-f005:**
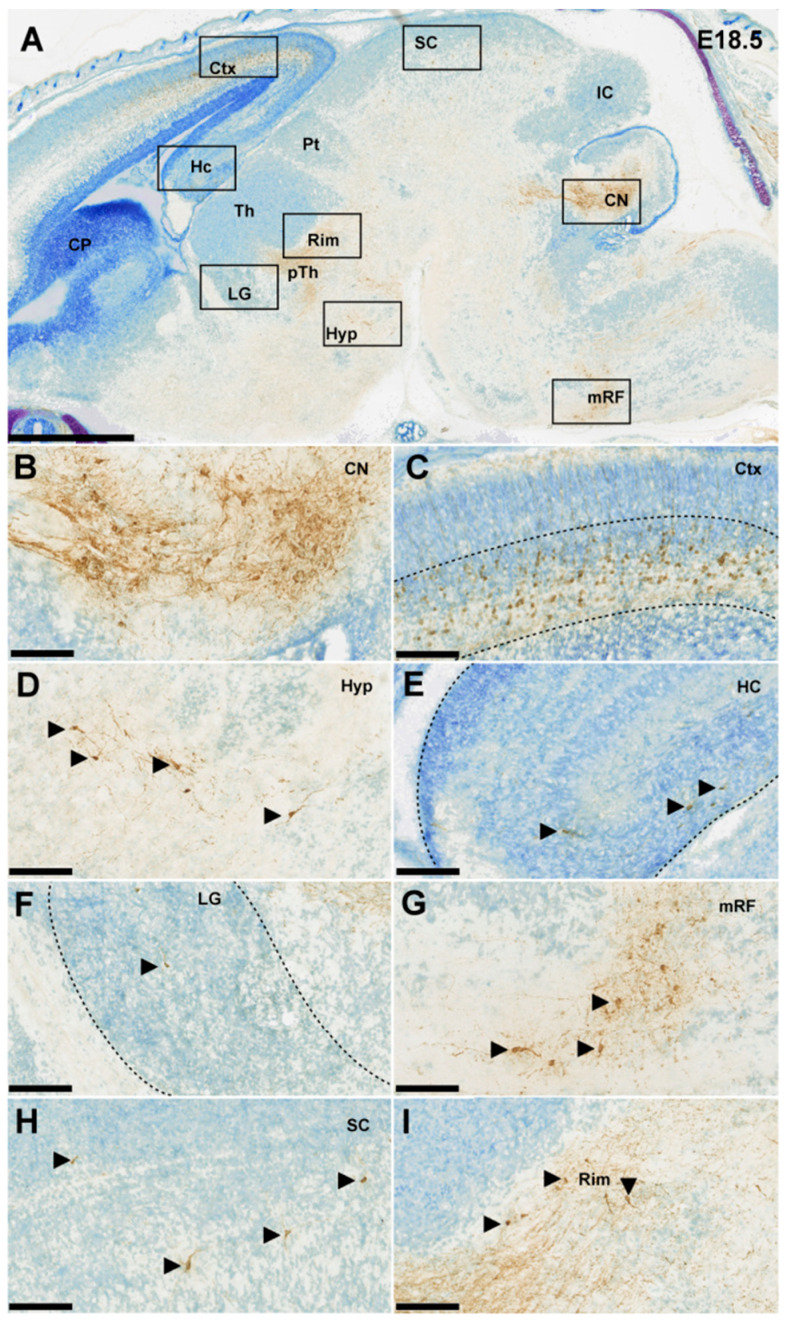
Ntsr1+ cells DAB-stained in an E18.5 embryonic mouse brain; counterstained with thionin. (**A**) Sagittal section showing all the regions in which Ntsr1 is expressed. Scale bar = 1000 µm (**B**–**H**) 20× zoom in pictures of the CN, the Ctx, the Hyp, the HC, the LG, the mRF and the SC, respectively. (**I**) 20× zoom in image of a region at the interface between thalamus and pTh with Ntsr1+ cells presumably originating from the Rim. Scale bar = 100 µm. CN = cerebellar nuclei, CP = caudate putamen, Ctx = cerebral cortex, HC = hippocampus, Hyp = hypothalamus, IC = inferior colliculus, LG = lateral geniculate nucleus, mRF = medial reticular formation, Pt = pretectum, pTh = prethalamus, SC = superior colliculus, Th = thalamus.

**Figure 6 cells-11-03800-f006:**
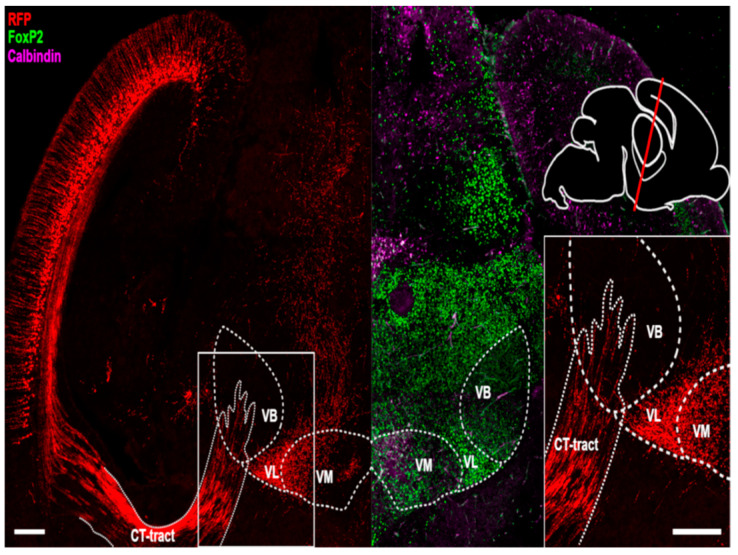
RFP^+^ cortical fibers entering the thalamus, specifically the ventrobasal thalamic nucleus, at E18.5. Coronal section of an E18.5 mouse brain stained for FoxP2 (in green), calbindin (magenta) and Red Fluorescent Protein (in red), with zoomed-in inset, showing the RFP^+^ corticothalamic fibers and another, separate, RFP^+^ fiber bundle, presumably originating from the cerebellar nuclei. Top right, schematic showing the angle and rostro-caudal level of the coronal slices. Scale bars: main image = 250 µm, inset = 100 µm. CT-tract = corticothalamic tract, VB = ventrobasal nucleus, VL = ventrolateral nucleus, VM = ventromedial nucleus.

**Figure 7 cells-11-03800-f007:**
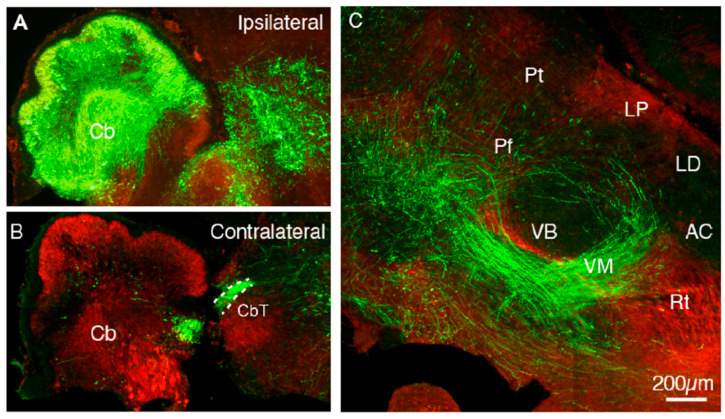
EYFP^+^ fibers of IUE transfected cerebellum are present in the thalamus at E18.5. (**A**,**B**) EYFP^+^ signal in the transfected ipsilateral cerebellum. (**C**) EYFP^+^ fibers in the thalamic complex show selective innervation of particular nuclei. Note that in this sagittal section several CbT-innervated nuclei are outside the field of view, i.e., on a different level of the medio-lateral axis. Cb = cerebellum; CbT = cerebellothalamic tract; Pt = pretectum; Pf = parafascicular nucleus; VB = ventrobasal nucleus; VM = ventromedial nucleus; Rt = reticular nucleus; AC = anterior complex; LD = lateral dorsal nucleus; LP = lateral posterior nucleus.

**Figure 8 cells-11-03800-f008:**
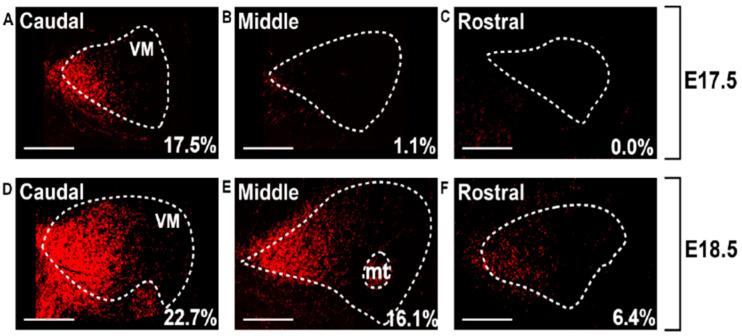
RFP^+^ fibers in embryonic thalamus of Ntsr1-Cre/AI14 mice progress rostral through the ventromedial nucleus. At three rostrocaudal levels the percentage of summed area occupied by above-threshold RFP-signal (pSAO) is reported for E17.5 (**A**–**C**) and E18.5 (**D**–**F**). Note that the indicated percentages are not equal to those reported in Figure 9 and Figure 10, since the latter represent the averages from bilateral VM of several sections.

**Figure 9 cells-11-03800-f009:**
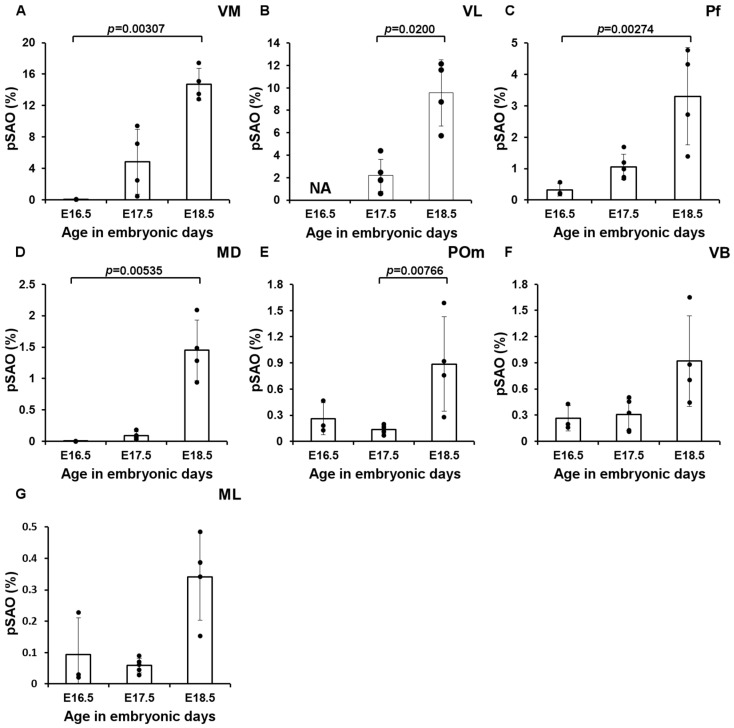
RFP signal quantification in the thalamic nuclei. Percentage of summed area occupied by above-threshold RFP-signal (pSAO). (**A**) VM (E16.5, n = 3, E17.5, n = 4, E18.5, n = 4), (**B**) VL (E16.5, n = 0, E17.5, n = 4, E18.5, n = 4) (**C**) Pf (E16.5, n = 3, E17.5, n = 5, E18.5, n = 4), (**D**) MD (E16.5, n = 2, E17.5, n = 5, E18.5, n = 4), (**E**) POm (E16.5, n = 3, E17.5, n = 5, E18.5, n = 4) (**F**) VB (E16.5, n = 3, E17.5, n = 5, E18.5, n = 4), (**G**) ML (E16.5, n = 3, E17.5, n = 5, E18.5, n = 4). Since the VL could not reliably be delineated at E16.5, as the border between VL and the intralaminar nuclei could not consistently be accurately delineated in all slices, we measured the pSAO in this nucleus at E17.5 and E18.5. Columns represent the mean; error bars represent ± SD; dots represent individual data points.

**Figure 10 cells-11-03800-f010:**
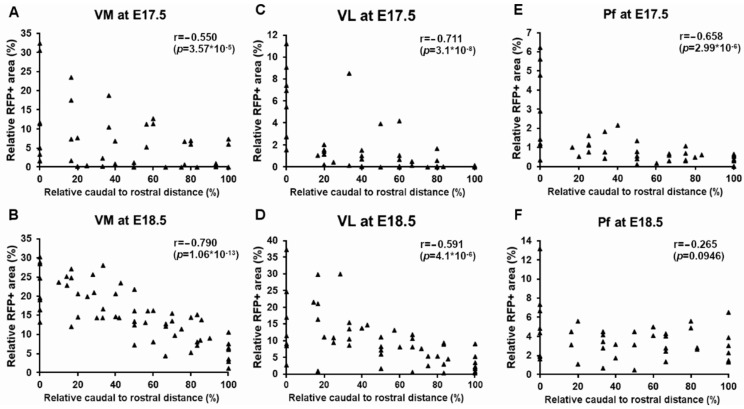
RFP signal quantification from caudal to rostral in the thalamic nuclei. The relative amount of RFP^+^ fibers (in %) is plotted against the relative rostral to caudal distance (in %) in VM, VL and Pf. (**A**,**B**) VM at E17.5 (n = 48) and E18.5 (n = 57), respectively, (**C**,**D**) VL at E 17.5 (n = 46) and E18.5 (n = 52), respectively, (**E**,**F**) Pf at E17.5 (n = 39) and E18.5 (n = 39), respectively. r = Spearman’s rho.

**Figure 11 cells-11-03800-f011:**
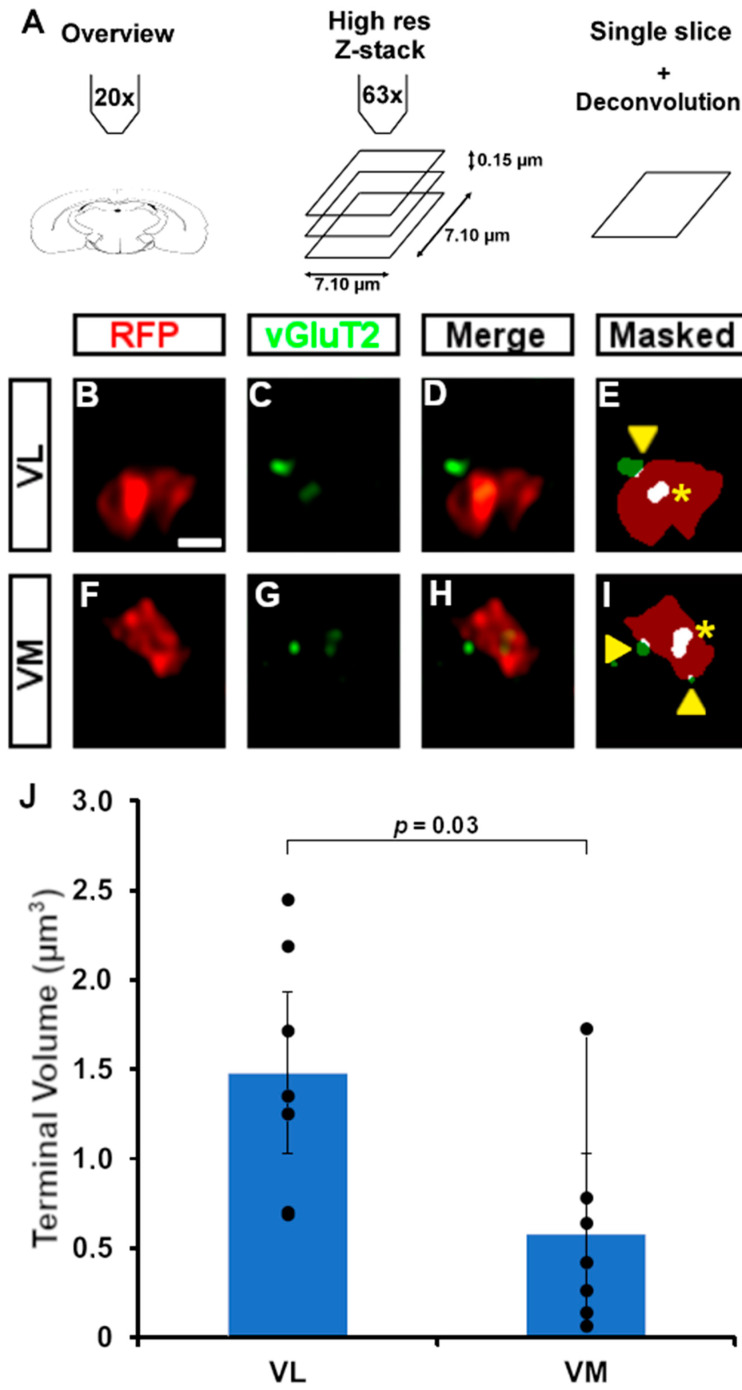
Putative boutons and colocalization with vGluT2. (**A**) Schematic overview of the workflow. After taking an overview image at 20×, we acquired z-stacks at 63×. Single optical slices were deconvolved and a user determined threshold was used after which we determined the overlap between fluorophores. RFP^+^ boutons were considered to colocalize with vGluT2 if the vGluT2-fluorescence overlapped completely in all dimensions. (**B**–**E**) Example of a deconvolved image of a putative bouton in VL, with RFP in red (**B**), vGluT2 in green (**C**), colocalization of RFP and vGluT2 (**D**), and the result after thresholding (**E**). Asterisks indicate complete overlap, arrowheads indicate partial overlap, the latter of which is not considered for further analysis. (**F**–**I**) as (**B**–**E**) for VM. (**J**) Terminal volumes (in µm^3^) in VL and VM. Scale bars: (**B**–**I**) = 1 µm.

**Figure 12 cells-11-03800-f012:**
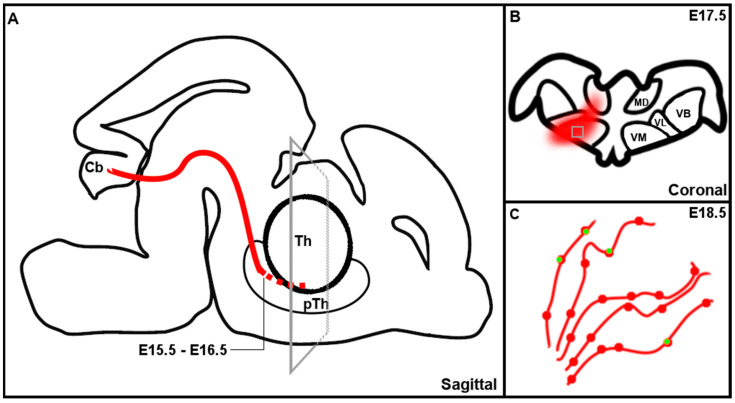
Schematic representation of the development of the cerebellothalamic tract. (**A**) Sagittal representation of the development of the cerebellothalamic tract (red). (**B**) Coronal representation of inset in A, showing schematic representation of the invasion of cerebellothalamic fibers (red) into the thalamus. (**C**) Zoom in of inset in (**B**), showing a schematic representation of the appearance of vGluT2 positive terminals (green) in the cerebellothalamic fibers (red) at E18.5. Th = thalamus, pTh = prethalamus, Cb = cerebellum, MD = mediodorsal nucleus, VB = ventrobasal nucleus, VL = ventrolateral nucleus, VM = ventromedial nucleus.

**Table 1 cells-11-03800-t001:** Primary antibodies.

Host Species	Antigen	Concentration	Manufacturer
Rabbit	Red fluorescent protein	1:1000	Rockland 600-401-379
Chick	Green fluorescent protein	1:200	Abcam ab13970
Guinea Pig	Vesicular glutamate transporter 2	1:500	MilliP AB2251
Goat	FoxP2	1:500	SC-21069
Chicken	Calbindin	1:500	Syn Sys 214006
Mouse	Calbindin D-28	1:100	Sigma CB955
Mouse	NeuN	1:1000	MilliP MAB377
Rabbit	Red fluorescent protein	1:1000	Rockland 600-401-379

**Table 2 cells-11-03800-t002:** Secondary antibodies.

Host Species	Target Species	Conjugate	Concentration	Manufacturer
Donkey	Rabbit	Cy3	1:400	Jackson 711-165-152
Donkey	Chick	FITC	1:200	Millipore AP194F
Goat	Rabbit	Biotin	1:1000	Jackson 111-065-144
Donkey	Guinea Pig	Cy5	1:200	Jackson 706-175-148
Donkey	Goat	Alexa488	1:200	Jackson 705-545-147
Donkey	Chicken	Cy5	1:200	Jackson 703-175-155
Goat	Mouse	Cy3	1:1000	Millipore AP124C
Donkey	Mouse	Alexa488	1:200	Jackson 715-545-150

**Table 3 cells-11-03800-t003:** Laser and filters per microscope per dye.

	Light-Sheet Ultramicroscope II		Opera Phenix HCS		Zeiss LSM700 Meta		
Dyes	Laser	Filters	Lasers	Filters	Lasers	Filters	Beamsplitter
Alexa 488	-	-	488 nm	500/50	488 nm	0/590	520
Cy3	561 nm	615/40	561 nm	570/30	555 nm	/640	600
Cy5	-	-	640 nm	650/60	633 nm	640/	630

**Table 4 cells-11-03800-t004:** Relative amount of RFP^+^ neurons (RFP/NeuN in %) per cerebellar nucleus and the relative amount of RFP^+^ neurons in each nucleus compared to the total amount of RFP^+^ CN neurons (RFP/totalRFP in %) for three mice. Data per mouse are averaged from 3 to 5 coronal sections.

	Mouse 1 (3 Sections)		Mouse 2 (5 Sections)		Mouse 3 (4 Sections)	
	RFP/NeuN in %	RFP/totalRFP	RFP/NeuN in %	RFP/totalRFP	RFP/NeuN in %	RFP/totalRFP
LCN	20.1	13.9	19.3	9.9	23.2	16.1
Int	47.7	72.2	56.8	80.2	50.4	74.3
MCN	21.1	14.0	10.7	9.9	13.4	9.6
Total	34.9	100	35.1	100	34.6	100

CN = cerebellar nuclei, Int = interposed nucleus, LCN = lateral cerebellar nucleus, MCN = medial cerebellar nucleus.

**Table 5 cells-11-03800-t005:** pSAO values (in %) of the investigated thalamic nuclei and the Z- and *p*-values of the differences between the age groups of each of these nuclei. Šidák-corrected *p*-values of 0.0127 were used for all nuclei, except VL for which a Šidák-corrected *p*-value of 0.017 was chosen. Significant differences are indicated in bold letters. For VB there was no post hoc test since the Kruskal–Wallis analyses showed no significant difference.

	E16.5	E17.5	E18.5	E16.5 vs. E17.5		E16.5 vs. E18.5		E17.5 vs. E18.5	
	pSAO	pSAO	pSAO	Z	*p*	Z	*p*	Z	*p*
VM	0.0480%	4.85%	14.7%	−1.38	0.167	−2.96	**0.00307**	−1.71	0.0881
VL	-	2.20%	9.56%	-	-	-	-	-	0.02
MD	0.00147%	0.0590%	1.45%	−1.26	0.207	−2.79	**0.00535**	−2.02	0.0431
Pf	0.330%	1.06%	3.30%	−1.60	0.111	−3.00	**0.00274**	−1.67	0.0940
ML	0.0930%	0.0586%	0.342%	−1.77	0.859	−2.15	0.0317	−2.25	0.0242
VB	0.263%	0.305%	0.919%	-	-	-	-	-	-
POm	0.0261%	0.137%	0.885%	0.836	0.403	−1.54	0.123	−2.67	**0.00766**

MD = mediodorsal nucleus, ML = midline nuclei, Pf = parafascicular nucleus, POm = posteriomedial nucleus, pSAO = percentage of summed area occupied by RFP-signal, VB = ventrobasal nucleus, VL = ventrolateral nucleus, VM = ventromedial nucleus. Bold number indicate significant differences.

## Data Availability

The data presented in this study are available upon request to the corresponding author.

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
