# Peer review of "Anatomical Development of the Cerebellothalamic Tract in Embryonic Mice"

_cells, 2022, doi:10.3390/cells11233800_

Round 1
Reviewer 1 Report
The manuscript titled Anatomical development of the cerebellothalamic tract in embryonic mice by Dumas et al. describes the progression of cerebellar connections to the thalamus from E15.5 through E18.5 using a combination of transgenic mice and in utero electroporation of cerebellum. By age E15.5 neurons originating in cerebellum reach the prethalamus then start to innervate thalamic nuclei after E16.5. Multiple techniques were used to rule out results of possible cerebellar thalamic connections originating from inputs associated with other brain areas. The results were clearly presented throughout the well written manuscript and the fluorescence quantification seemed appropriate. The framing of results with how abnormalities in cerebellar connections/communication to forebrain structures being a hallmark feature of neurological disorders during development maps nicely onto the importance of this work. The authors should be commended for their rigorous investigation of the development of the cerebellothalamic tract during this early stage of development. I have only minor comments.
1. Are Figure 1E and Table 4 both necessary? Both show the same information, RFP/NeuN% for cerebellar neurons. Aside for the information being redundant, the numbers in the table and bar graphs do not match up. Was this data from different mice? I suggest the authors either remove Figure 1E, or double check to make sure the numbers in Figure 1E match the numbers from Table 4 if the bar graph is kept in Figure 1. Also these data are from much older mice than the focus of the study. It should be stated somewhere that it was assumed (or not) that the authors expected similar findings in cerebellar neurons from younger mice.
2. The authors should help orient the reader to where in the embryonic brain these sections are located. It can be assumed that these sections are relatively caudal based on the results presented later in the paper but it would be nice to at very least have a smaller version of Figure 10A to show the rostral/caudal extent of these projections in prethalamus. Also there are many other RFP fibers in the section from Figure 2A, the location of those fibers should be outlined and briefly mentioned.
3. Show a more zoomed out version of the dorsal perspective for Figure 3A and if possible show an additional (lateral) zoomed out perspective to help the reader see the spatial relationship between structures with RFP. It would be helpful to also have a matching schematic for each orientation (similar to Figure 10A) showing the extent of these fibers. The authors should also include smaller versions of Figure 10A to help the reader orient to the progression of fibers into the thalamus from E17.5 to E18.5 for 3B, 3D and 3F. The visualization in the other parts of the figure are clear and convincing.
4. Figure 4 is very clear and effective due to the zoomed out nature of Figure 4A. Providing this level of perspective in many of the other figures would benefit the presentation of those figures in the same way.
5. The structural outlines in Figure 5A are not consistent with the rest of the other figures and look relatively sloppy. I suggest the authors make the solid lines into dotted lines (consistent with all the other figures) and simplify the details.
6. Figures 6A and 6B should either be zoomed out to show the distribution of EYFP+ fibers throughout the brain, and/or, add more labels to mark the fibers outside of the cerebellum. For instance, CbT = cerebellothalamic tract and SC = superior colliculus are both in the figure legend but neither structure is indicated in the figure. Would be interesting to see other medial/lateral sections to show the CbT-innervated nuclei and extent of EYFP+ expression throughout the cerebellum.
7. Quantification of pSAO values across ages is a valuable contribution of this work. Authors should include at least 1 example into Figure 7 (with either all data from each animal merged at the different ages or individual examples at different ages) of the expression pattern for a given area with the specific area outlined similar to previous figures. The same goes for Figure 8 to show the caudal to rostral extent.
8. The images in Figure B-I need to be higher resolution and the indicators of overlap (asterisks) and partial overlap (arrowheads) are not visible in the current figure.
9. The authors should label CbT on the schematic in Figure 10A. If smaller versions of Figure 10A are added to previous figures there should be callouts to those figures along with the indications of the rostral projection extent at each embryonic age. Having an additional smaller insert of a dorsal view of the developmental brain indicating the medial/lateral location of the main part of Figure 10A would be helpful.
Author Response
Reviewer 1
The manuscript titled Anatomical development of the cerebellothalamic tract in embryonic mice by Dumas et al. describes the progression of cerebellar connections to the thalamus from E15.5 through E18.5 using a combination of transgenic mice and in utero electroporation of cerebellum. By age E15.5 neurons originating in cerebellum reach the prethalamus then start to innervate thalamic nuclei after E16.5. Multiple techniques were used to rule out results of possible cerebellar thalamic connections originating from inputs associated with other brain areas. The results were clearly presented throughout the well written manuscript and the fluorescence quantification seemed appropriate. The framing of results with how abnormalities in cerebellar connections/communication to forebrain structures being a hallmark feature of neurological disorders during development maps nicely onto the importance of this work. The authors should be commended for their rigorous investigation of the development of the cerebellothalamic tract during this early stage of development. I have only minor comments.
Are Figure 1E and Table 4 both necessary? Both show the same information, RFP/NeuN% for cerebellar neurons. Aside for the information being redundant, the numbers in the table and bar graphs do not match up. Was this data from different mice? I suggest the authors either remove Figure 1E, or double check to make sure the numbers in Figure 1E match the numbers from Table 4 if the bar graph is kept in Figure 1.
Answer: We thank the reviewer for picking up this omission between data represented in Figure 1E and Table 4. To avoid redundancy in showing RFP/NeuN%, we deleted Figure panel 1E and updated the numbers reported in Table 4.
Also these data are from much older mice than the focus of the study. It should be stated somewhere that it was assumed (or not) that the authors expected similar findings in cerebellar neurons from younger mice.
Answer: We indeed included data from older mice in Table 4 compared to the focus of the study. We did perform RFP/NeuN% quantification in late embryonic stages, but the population of NeuN+ cells in the CN region at these ages is not representative for the number of CN neurons, since cerebellar cortical neurons, which at that stage are potentially NeuN+, may migrate through this region. We do assume that the ratio between RFP+ cells between LCN/IN/MCN is stable, since CN glutamatergic neuronal proliferation already completes at E12.5 and migration to the nuclear transitory zone before E16.5 (Fink et al 2006, J Neurosci). This is explained in lines 338-343 on page 8.
The authors should help orient the reader to where in the embryonic brain these sections are located. It can be assumed that these sections are relatively caudal based on the results presented later in the paper but it would be nice to at very least have a smaller version of Figure 10A to show the rostral/caudal extent of these projections in prethalamus.
Answer: We agree with the reviewer on the importance of helping the reader to orient in the rostro-caudal axis. We have included a smaller version of the schematic Figure 10A into Figures 2C, 3B, 3D, 4A and 6.
Also there are many other RFP fibers in the section from Figure 2A, the location of those fibers should be outlined and briefly mentioned.
Answer: We have now outlined RFP fibers in Figure 2A and reported this in the legend.
Show a more zoomed out version of the dorsal perspective for Figure 3A and if possible show an additional (lateral) zoomed out perspective to help the reader see the spatial relationship between structures with RFP. It would be helpful to also have a matching schematic for each orientation (similar to Figure 10A) showing the extent of these fibers.
Answer: We understand the comment of the reviewer. We have tested the visibility of the CbT in a more zoomed out version in the dorsal and lateral views. Due to the relatively high density of RFP+ neurons in layer 6 of the cerebral cortical regions (e.g. see figures 5C and 6 of the revised manuscript), the visibility of the CbT in the diencephalic regions – the main focus of the paper – is too limited. One option is to provide an online video montage in which the scan is rotating in 3D, but we are not convinced that this will provides additional anatomical data compared to the histological data represented in Figure 4. In case the reviewer and/or editor prefers a video montage in 3D, we are more than willing to provide the best possible version for the online supplementary material.
The authors should also include smaller versions of Figure 10A to help the reader orient to the progression of fibers into the thalamus from E17.5 to E18.5 for 3B, 3D and 3F. The visualization in the other parts of the figure are clear and convincing.
Answer: We have now included a smaller version of Figure 10A in panels 3B and 3D. Panel 3F from the old version is now Figure 4A, which also includes the same schematic.
The structural outlines in Figure 5A are not consistent with the rest of the other figures and look relatively sloppy. I suggest the authors make the solid lines into dotted lines (consistent with all the other figures) and simplify the details.
Answer: To increase consistency and clarity, we have redrafted this figure panel (old version figure 5A, new version Figure 6) according to the reviewers comments.
Figures 6A and 6B should either be zoomed out to show the distribution of EYFP+ fibers throughout the brain, and/or, add more labels to mark the fibers outside of the cerebellum. For instance, CbT = cerebellothalamic tract and SC = superior colliculus are both in the figure legend but neither structure is indicated in the figure. Would be interesting to see other medial/lateral sections to show the CbT-innervated nuclei and extent of EYFP+ expression throughout the cerebellum.
Answer: We have now included the label ‘CbT’ into the relevant figure panel (Figure 7B in the revised manuscript). There were no other sections of adequate quality available to include.
Quantification of pSAO values across ages is a valuable contribution of this work. Authors should include at least 1 example into Figure 7 (with either all data from each animal merged at the different ages or individual examples at different ages) of the expression pattern for a given area with the specific area outlined similar to previous figures. The same goes for Figure 8 to show the caudal to rostral extent.
Answer: We agree with the reviewer that example images for the pSAO values are an important addition. We have now included examples for the ventromedial nucleus at two different ages (E17.5 and E18.5) and three different rostro-caudal levels in a separate figure. In the new version of the manuscript this is Figure 8.
The images in Figure B-I need to be higher resolution and the indicators of overlap (asterisks) and partial overlap (arrowheads) are not visible in the current figure.
Answer: We have increased the contract of these images (Figure 11B-I in the revised manuscript) to maximize the resolution of our images and made sure the indicators for overlap and partial overlap are now visible.
The authors should label CbT on the schematic in Figure 10A. If smaller versions of Figure 10A are added to previous figures there should be callouts to those figures along with the indications of the rostral projection extent at each embryonic age. Having an additional smaller insert of a dorsal view of the developmental brain indicating the medial/lateral location of the main part of Figure 10A would be helpful.
Answer: We thank the reviewer for these helpful suggestions and have adapted this schematic representation of the developing brain (Figure 12 in the revised manuscript). We also included the smaller versions in other figure panels, as indicated previously in this rebuttal.
Reviewer 2 Report
In manuscript, Dumas et al focused on the ontogeny of cerebellothalamic tract (CbT) in embryonic mice. They utilized Ntsr1-Cre/Ai14 mice as model to investigate the development of CbT using imaging-based approaches. Meanwhile, they also performed in utero electroporation of pCAG-EYFP plasmid into fourth ventricle on wild-type mice to validate. Dumas et al observed interesting phenotypes correlated to CbT development during embryonic mice development from E14.5 to E18.5. However, one of my major concerns is that the data shown here are almost all imaging based. Is that possible to include more evidence from different angles? For example, RNA and protein level analysis following targeted cell sorting with more diverse markers. If Ntsr1 promoter is not specific enough for CbT labeling, I will suggest more evidence from different angles. The following are a few specific points:
1. In figure 1E, the data shown here seems from one slice per mouse. If that is the case, I suggest including more slices from each mouse and then do statistical analysis.
2. In “results” section, I suggest use subsections and subtitles to conclude each research points.
Author Response
Reviewer 2
In manuscript, Dumas et al focused on the ontogeny of cerebellothalamic tract (CbT) in embryonic mice. They utilized Ntsr1-Cre/Ai14 mice as model to investigate the development of CbT using imaging-based approaches. Meanwhile, they also performed in utero electroporation of pCAG-EYFP plasmid into fourth ventricle on wild-type mice to validate. Dumas et al observed interesting phenotypes correlated to CbT development during embryonic mice development from E14.5 to E18.5. However, one of my major concerns is that the data shown here are almost all imaging based. Is that possible to include more evidence from different angles? For example, RNA and protein level analysis following targeted cell sorting with more diverse markers. If Ntsr1 promoter is not specific enough for CbT labeling, I will suggest more evidence from different angles.
Answer: We understand that the reviewer would like to see RNA and protein level analysis following targeted cell sorting with more diverse markers. We frankly think that this is beyond the scope of the current manuscript. We do not share the reviewers concern about the current dataset being solely from imaging data. Our qualitative and quantitative assessments of the developing CbT in the embryonic mouse brain are robust and provide novel insight in the ontogeny and maturation of the cerebello-thalamic connection.
The following are a few specific points:
In figure 1E, the data shown here seems from one slice per mouse. If that is the case, I suggest including more slices from each mouse and then do statistical analysis.
Answer: We thank the reviewer for raising our attention to the data shown in this figure panel. We have removed this panel (per request of reviewer 1) and provided all data from the three adult mice in Table 4, including the number of utilized coronal sections per mouse. Since the point of these data is to merely report the proportion of RFP+ cells per cerebellar nuclei and the number of included mice is limited, we choose to not include statistical analysis.
In “results” section, I suggest use subsections and subtitles to conclude each research points.
Answer: We have now included subsections and subtitles.